
# Transferability of data-driven models to predict urban pluvial flood water depth in Berlin, Germany

Omar Seleem[1], Georgy Ayzel[1], Axel Bronstert[1], and Maik Heisterman[1]

[1]Chair of Hydrology, Institute of Environmental Science and Geography, University of Potsdam

**Correspondence:** Omar Seleem (seleem@uni-potsdam.de)

**Abstract.** Data-driven models have been recently suggested to surrogate computationally expensive hydrodynamic models to map flood hazards. However, most studies focused on developing models for the same area or the same precipitation event. It is hence not obvious how transferable the models are in space. This study evaluates the performance of a convolutional neural network (CNN) based on the U-Net architecture and the random forest (RF) algorithm to predict flood water depth, the models' transferability in space and performance improvement using transfer learning techniques. We used three study areas in Berlin to train, validate and test the models. The results showed that (1) the RF models outperformed the CNN models for predictions within the training domain, presumable at the cost of overfitting; (2) the CNN models had significantly higher potential than the RF models to generalize beyond the training domain; and (3) the CNN models could better benefit from transfer learning technique to boost their performance outside training domains than RF models.

## 1 Introduction

Urbanization increases the frequency and severity of extreme urban pluvial flood events (Skougaard Kaspersen et al., 2017). Therefore, it is crucial to quantify the flood water depth and extent due to pluvial flooding in urban environments. While 2-dimensional hydrodynamic models are effective and robust in estimating urban floodwater depth, they are difficult to scale due to prohibitive computational costs (Costabile et al., 2017). Data-driven models are raising as a surrogate might overcome the limitations of the computationally expensive numerical models (Hou et al., 2021; Guo et al., 2021; Löwe et al., 2021; Guo et al., 2022; Bentivoglio et al., 2022). They do not simulate the physical process of runoff generation and concentration, but find patterns between the input and output data. The model's accuracy depends on the amount, quality and diversity of the available data. They could predict water depth with a sufficient level of accuracy within seconds. Consequently, they are a helpful tool that can support decision-makers with a real-time forecast.

Data-driven models used to address urban pluvial flood hazards in the literature can be grouped into models that use only rainfall input to map flood hazards (Hou et al., 2021; Hofmann and Schüttrumpf, 2021), and models that account for the topographic characteristics of the urban landscape (Löwe et al., 2021; Guo et al., 2022). The former group interpolates the flood response between rainfall events that were used to train the model and hence can only predict flood hazards within the training domain while the latter has the potential to generalize and make accurate predictions outside the training domain (Bentivoglio et al., 2022).



Point-based data-driven models such as the random forest (RF) algorithm have been widely used in the literature to map susceptibility for pluvial flooding (Lee et al., 2017; Chen et al., 2020; Zhao et al., 2020; Seleem et al., 2022). RF models outperformed convolutional neural networks (CNN) to map flood susceptibility in Berlin at various spatial resolutions, and showed promising results outside the training domain (Seleem et al., 2022). Hou et al. (2021) trained RF and K-nearest neighbour (KNN) algorithms to predicate urban pluvial flood water depth using only the rainfall characteristics as inputs, and Zahura et al. (2020) trained a RF model to predict flood water depth in an urban coastal area using three topographic predictive features. However, both studies evaluated the model performance inside the training domain only. The algorithm performance to map urban pluvial flood hazards using different topographic characteristics of the urban area and its ability to generalize to other areas than the training domain have not been systematically investigated in the literature, yet.

CNNs have recently demonstrated the potential to map urban pluvial flood susceptibility (Zhao et al., 2020, 2021; Seleem et al., 2022) and flood hazard (Löwe et al., 2021; Guo et al., 2022).They are designed to extract spatial information from the input data and to handle image (raster) data without an unwarranted growth in the model complexity. Löwe et al. (2021) trained a CNN model based on the U-Net architecture (Ronneberger et al., 2015) to predict urban pluvial flood water depth. They divided the city into a grid, used part of it for training and the rest for testing. The testing areas were close to or surrounded by training areas which guaranteed that the testing dataset had minimal diversity from the training dataset. Guo et al. (2022) used four topographic predictive features and one precipitation event to train a CNN model. The model performed well outside the training domain for the same precipitation event used to train the model.

Deep learning uses transfer learning techniques to mitigate the problem of insufficient training data (Tan et al., 2018). Zhao et al. (2021) applied transfer learning techniques to map urban pluvial flood susceptibility using the LeNet-5 network architecture. A model that was trained on a certain part of the city (pre-trained model) performed poorly outside the training domain. A transferred model trained by freezing the pre-trained model weights and allowing only a few weights to be re-trained using a few new training data from the new area improved the model performance. The transferred model used the knowledge learned from the pre-trained model and outperformed a model that was only trained for the new area. These techniques have not yet been investigated for predicting flood water depth or for shallow machine learning algorithms such as RF.

In summary, deep learning was consistently superior to shallow machine learning in literature but recent studies showed the contrary (Seleem et al., 2022; Grinsztajn et al., 2022). However, shallow machine learning algorithms have not been systematically challenged in terms of transferability for urban flood modelling. A data-driven model that generalizes outside the training domain is still a major challenge in literature (Bentivoglio et al., 2022). While previous studies tried to examine the transferability of CNN in space to predict flood water depth under certain limitations (Löwe et al., 2021; Guo et al., 2022) and use transfer learning techniques to improve the CNN performance outside the training domain to map flood susceptibility (Zhao et al., 2021), such efforts have been examined neither for RF models nor for surrogates of physical numerical 2D hydrodynamic models. It is not obvious how transfer learning techniques could improve the data-driven model performance and be a useful tool to overcome the limitations of applying computationally expensive 2D hydrodynamic models to a big region. In this study, we address the following research questions:



(1) How does the performance of RF and CNN models in predicting urban pluvial flood water depth compare inside and outside the training domain?

(2) Can transfer learning techniques improve the model performance outside the training domain and thus help to overcome the issue of limited training data?

## 2 Methodology

### 2.1 Study design

The overall design of this study was as follows: firstly, we selected three areas (Figure. 1) that have frequently been flooded in the last decades based on a flood inventory (Seleem et al., 2022) gathered between 2005 and 2017. 2D hydrodynamic simulations were carried out in these areas. Then, the precipitation depth, topographic predictive features and water depth from the 2D hydrodynamic simulations were used to prepare the training, validation and testing datasets. We randomly selected 10000 images (raster with spatial extent 256 × 256) and 10 % of the available data ( number of pixels within the training domain × number of training precipitation events) to develop both the U-Net and RF models respectively. We split the data into training (60 %), validation (20 %) and testing (20 %) datasets. The validation dataset were used to estimate the optimal hyperparameter combinations. The testing dataset included data from three precipitation events (50, 100, and 140 mm) which were not included in the training and validation datasets. Next, we defined six combinations of training and testing datasets as shown in Table 1,and evaluated the model performance inside each training domain and the models' spatial transferability to other testing domains, hence we evaluated the transferability between precipitation events (at the same training domains) and the transferability in space between study areas. Afterwards, we selected the best hyperparameter combinations for the data-driven model that best fit the training dataset. Finally, we investigated whether the learned knowledge from the pre-trained models can improve urban flood hazard mapping outside the training domain using transfer learning techniques and which predictive features are mostly influencing the model predictions.

**Table 1.** Examined training data combinations to train the data-driven models.

| Training domain | Testing domain | Training domain | Testing domain |
|---|---|---|---|
| SA0 | SA0*, SA1, & SA2 | SA0 & 1 | SA0*, SA1*, & SA2 |
| SA1 | SA0, SA1*, & SA2 | SA0 & 2 | SA0*, SA1, & SA2* |
| SA2 | SA0, SA1, & SA2* | SA1 & 2 | SA0, SA1*, & SA2* |

* refers to testing the model with precipitation events that were not included in the training dataset.



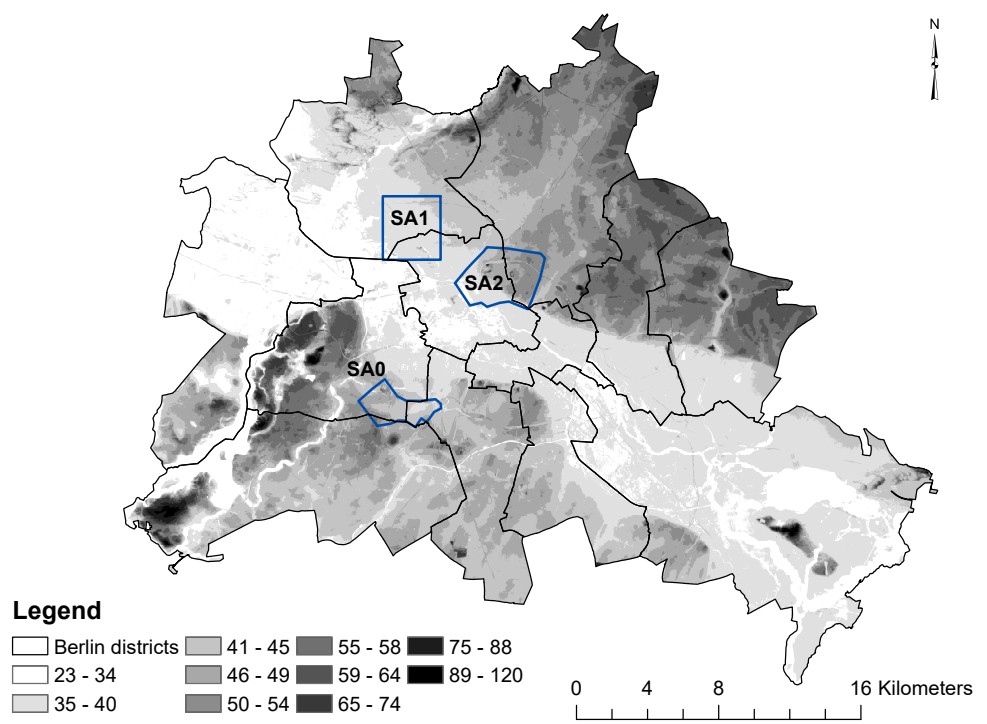

**Figure 1.** The three study areas in Berlin.

## 2.2 Study area and hydrodynamic model

Berlin is the capital of Germany and has around 3.6 million inhabitants. The city has a relatively flat topography (Seleem et al., 2022) and has an oceanic climate (Köppen: Cfb) (Peel et al., 2007). The average annual precipitation is around 570 mm (Berghäuser et al., 2021). Heavy summer precipitation caused several urban pluvial floods in the last decades, for example, the

170 mm precipitation depth event on the 29 [th] and 30 [th] of June 2017 (Berghäuser et al., 2021). The selected study areas are between 6, 11, and 12 km$^2$. Seleem et al. (2021) showed that SA0 has large deep topographic depressions where flood water tends to accumulate, while flood water spill outside the topographic depressions after a certain precipitation depth threshold in SA2.

   The maximum water depths were obtained from TELEMAC-2D (Galland et al., 1991) hydrodynamic simulations (for SA0

and SA2) performed by (Seleem et al., 2021). We performed additional simulations for SA1 using the same model setup. We used the finite volume scheme to solve the shallow water equations over non-structured triangular grids (1 m maximum side length). The simulations were carried out using one-hour duration precipitation events with precipitation depths ranging from 20 to 150 mm (10 mm increments). We used the SCS-CN method (Cronshey, 1986) to estimate excess runoff. For more information about the model setup, please see (Seleem et al., 2021).

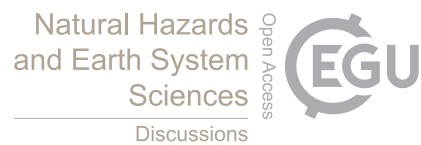

## 2.3 Predictive features

While data-driven models do not "understand" the physical processes of runoff generation and concentration, they are designed to detect relationships between input and target variables (Grant and Wischik, 2020), in this case simulated inundation depth. Therefore, predictive features should represent the surface characteristics of the study area which could inform the model of governing hydrological and hydrodynamic patterns. Table 2 shows the selected 12 predictive features that we considered potentially relevant for mapping urban floods and their description. The topographic predictive features were generated from a digital elevation model (DEM) with a 1 x 1 m pixel size which is openly available to download for the entire city of Berlin (ATKIS, 2020).

## 2.4 Models

### 2.4.1 U-Net

The application of CNNs for mapping urban flood hazards is still rare in the literature (Löwe et al., 2021). This study adopted the U-Net architecture (Ronneberger et al., 2015) as shown in Figure 2. The U-Net architecture showed a good performance to predict water depth in the literature (Löwe et al., 2021; Guo et al., 2022). The model input is a terrain raster with 13 image channels (13 channels represent the predictive features ) and the output is the resulting water depth at the surface. The U-Net architecture belongs to encoder/decoder architectures. The encoder follows the typical architecture of a convolutional neural network and uses pooling to downscale the spatial resolution, while the decoder uses upsampling to upscale the learned patterns. Skip connections concatenate the output of each encoder layer to its corresponding decoding layer to provide the spatial information (Srivastava et al., 2015).

We applied LeakyReLU with an activation threshold of 0.2 to all layers except the output layer (Maas et al., 2013; Löwe et al., 2021; Guo et al., 2022) and adaptive moment estimation (Adam; Kingma and Ba, 2014) to update and optimize the network weights. We used average pooling because it showed better performance than maximum pooling (Löwe et al., 2021), and added a batch normalization layer after each convolutional layer to stabilize and speed up the training process (Ioffe and Szegedy, 2015; Santurkar et al., 2018). A drop-out strategy was implemented with a rate of 0.5 to the convolutional layers (Löwe et al., 2021; Seleem et al., 2022), and early stopping to prevent overfitting (Prechelt, 1998). We used a batch size of 10 and the mean squared loss as a loss function to train the models (Löwe et al., 2021).

The success of CNN relies on finding a suitable architecture that fits a given task (Miikkulainen et al., 2019). Therefore, we varied three parameters similar to (Löwe et al., 2021) to obtain the most suitable network architecture, namely the network depth (i.e. number of encoding and decoding blocks) (varied between 3 and 4), number of filters in the first convolutional layer (varied between 16, 32 and 64) and the size of the kernels in the convolutional layers (varied between 3, 5, and 7). Using a deeper network and more filters increases the number of parameters and the computational expense. Moreover, using a larger kernel size allows the network to perform spatial aggregation on a larger region, again, however, at increasing computational cost.



**Table 2.** Spatial predictive features used to train the data-driven models.

| Predictive feature | Data adjustment | Description |
|---|---|---|
| Altitude | Normalized to [0,1] | Surface elevation is important for flood hazard mapping because runoff tends to accumulate at low elevation (Zhao et al., 2020; Seleem et al., 2021; Löwe et al., 2021; Seleem et al., 2022) |
| Slope | Normalized to [0,1] | Slope impacts the runoff velocity and the available time for infiltration (Rahmati et al., 2016) |
| Aspect | Scaled to [-1,1] | Aspect indicates the flow direction. We used the cosine and sine of aspect as two separate predictive features to deal with the cyclic behaviour of flow direction (Löwe et al., 2021). (Löwe et al., 2021; Seleem et al., 2022) found that aspect was the most important predictive feature for mapping urban floods using CNNs. |
| TWI | Normalized to [0,1] | Topographic wetness index was proposed by (Kirkby, 1975). It indicates the geotechnical wetness level and is being used to identify urban flood-prone areas (Jalayer et al., 2014; Seleem et al., 2021). |
| Curvature | Normalized to [-1,1] | Depending on the curvature value, the surface is flat, concave or convex. (Guo et al., 2021; Löwe et al., 2021) used it to predict urban flooding using data-driven models. |
| SDepth | Normalized to [0,1] | Depth of topographic depression impacts the volume of excess runoff that can be accumulated in it (Zhang and Pan, 2014; Seleem et al., 2021, 2022; Löwe et al., 2021). |
| FLACC | Normalized to [0,1] | Flow accumulation indicates the number of pixels draining into a certain pixel. We used the upper cutoff at 250 ha because very large values represent natural streams (Löwe et al., 2021). |
| TPI | Normalized to [-1,1] | Topographic position index is defined as the difference between the pixel elevation and the mean elevation of the surrounding pixels (Lei et al., 2021). A positive value denotes that the pixel is higher than the neighbouring pixels while a negative value indicates that the pixel is lower that the neighbouring pixels and a zero value represents flat areas (Weiss, 2001). |
| CN | Normalized to [0,1] | Curve number is an empirical parameter that is computed using land-cover and soil hydrologic group (Cronshey, 1986). It is used to estimate the direct runoff. We used the CN map produced by (Seleem et al., 2021). |
| Roughness | Normalized to [0,1] | Roughness impacts the excess runoff flow over the surface. We used the Manning roughness coefficient map produced by (Seleem et al., 2021). Buildings were defined by a high roughness coefficient similar to the TELEMAC - 2D model setup (Seleem et al., 2021). |
| DEML | Normalized to [0,1] | It is computed as the difference between the elevation of a pixel and the focal mean of elevation within 100 m radius. Urban pluvial floods occur on a small spatial scale (< 1 km) and are connected to the local variation in elevation (Löwe et al., 2021). |
| Precipitation depth | Normalized to [0,1] | We used one-hour duration precipitation events with precipitation depths ranging from 20 to 150 mm (10 mm increments) (Seleem et al., 2021). |




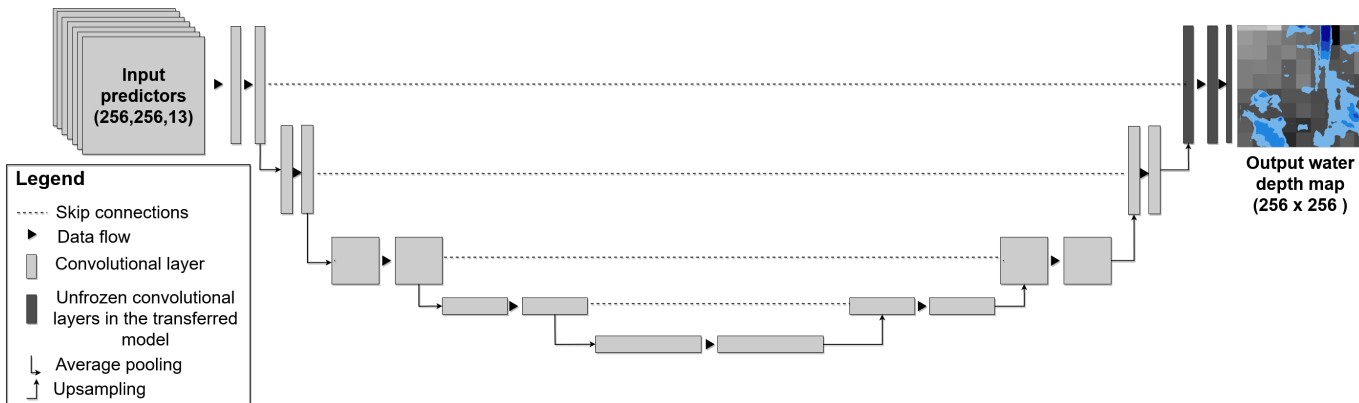

**Figure 2.** Schematic diagram of the applied U-Net architecture for a network of depth = 4 (4 blocks of encoder and decoder). The transferred model obtained the weights from the pre-trained model except the weights in the last decoder block (black colour). Then, the new training data was used to train the remaining untrained weights.

We implemented an input image size of $256 \times 256$ pixels ($1 \times 1$ m spatial resolution). Löwe et al. (2021) used the same image size but with a 5 m spatial resolution. We understand that this image size may be not sufficient to fully capture urban watersheds or topographic depressions. Then again, the selected study areas are small (area ranges from 6 to 12 km$^2$). We also used 12 predictive features to guarantee that the input data are well representing both the terrain and hydrological characteristics. The predictive features were calculated for the whole city and hence the calculated rasters consider the characteristics of the upstream urban catchment. Finally, training models with larger images is also limited by the memory of the graphic card.

### 2.4.2 Random forest

The random forest (RF) is a decision tree algorithm that was proposed by (Breiman, 2001). It solves both classification and regression problems by combining several randomized decision trees and averaging their predictions. RF divides the training data into several sub-datasets. Then, a tree model is developed for each dataset. Finally, a prediction is determined based on the majority result of the decision trees as shown in Figure 3. This approach intends to prevent overfitting (Biau and Scornet, 2016).

It is well known that RF perform relatively well with default hyper-parameter values. Still, hyper-parameter tuning may improve model performance (Probst et al., 2019). This study used the default values for the hyper-parameters such as the minimum number of samples in a node and the maximum depth of each tree in the sklearn.ensemble.RandomForestRegressor (Pedregosa et al., 2011), and varied the number of trees in the forest (between 10, 100, 200 and 300) (Zahura et al., 2020). Finally, an increasing number of training data points increases the training time and the model size dramatically. We used 10 % of the available training data (number of pixels within the training domain × number of training precipitation events) to train the RF model for all the simulations carried out in the study. We also tried to use larger portions of the training data, but without a significant improvement in model performance.





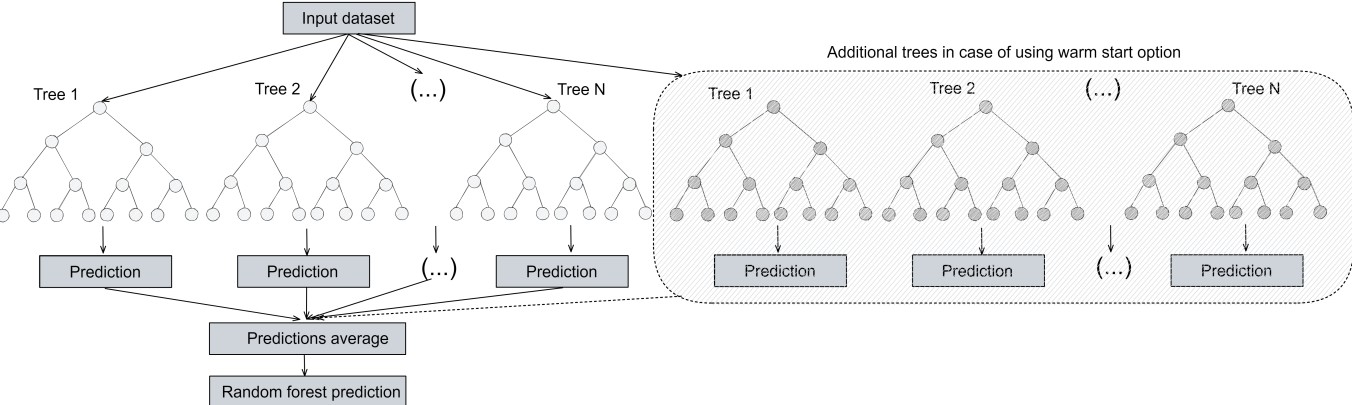

**Figure 3.** Schematic diagram of the random forest algorithm and the additional trees that are added to the model in case of a warm start. The additional trees are trained using the new training data while the old trees ( from the pre-trained model) remain unchanged.

## 2.5 Transfer learning

The transfer learning technique is a vital tool in deep learning to overcome the problem of insufficient training data (Tan et al., 2018). It is based on the idea that a model is firstly trained for a certain task (called the pre-trained model). Then, a new model is implemented (the transferred model) where some of its layers are frozen (they use the same weights from the pre-trained model) and the remaining layers (weights) are trained using new training data and/or an new task. This technique hence extend the application of data-driven models outside the training domain of the pre-trained model. It also reduces the training time because of the reuse of the weights from the pre-trained model. In this study, we froze all the layers in the U-Net model except the layers in the last decoding block which were then re-trained using new training data (see Figure 2) (Adiba et al., 2019).

The majority of shallow machine learning algorithms do not support transfer learning techniques because training the model is always fast and not complicated. However, RF offers adding more trees to the forest to be fitted using a new training dataset which means a model can be trained (pre-trained model) then new trees can be added to the forest and trained using the new training data ( transferred model) without changing the trees in the pre-trained model as shown in Figure 3.

## 2.6 Performance evaluation

The models' performance was assessed based on predicting water depth and inundation extent. For computing the performance indices, we compared the water depth and extent obtained from the TELEMAC-2D model to the results of the competing data-driven models. Table 3 gives an overview of performance metrics. We computed other indices like balanced accuracy, mean absolute error and the total flooded area ratio. However, we found that root mean square error (RMSE), Nash Sutcliffe efficiency (NSE) and critical success index (CSI) are well representing the model performance. A 10 cm threshold was applied for the CSI calculation.





**Table 3.** Performance indices used to evaluate the models' predictions. The $y_i$ and $\hat{y}_i$ denote the water depth from the TELEMAC-2D model and the data-driven model respectively. $\overline{\hat{y}_i}$ is the average of water depths from the data-driven model. Hits, misses and false alarms are estimated by the contingency table.

| Index | Equation | Range | Description |
|---|---|---|---|
| RMSE | $\sqrt{\frac{1}{n}\sum_{i=1}^{n}(y_i - \hat{y}_i)^2}$ | $[0, \infty]$ | Root mean square error measures the difference between the predicted and observed values. The optimal RMSE is zero. |
| NSE | $1 - \frac{\sum_{i=1}^{n}(y_i - \hat{y}_i)^2}{\sum_{i=1}^{n}(y_i - \overline{\hat{y}_i})^2}$ | $[-\infty, 1]$ | Nash Sutcliffe efficiency shows how well the observed values are predicted by the model (Nash and Sutcliffe, 1970). The optimal NSE value is one. |
| CSI | $\frac{hits}{hits+misses+falsealarms}$ | $[0, 1]$ | Critical success index is a binary index calculated based on pixel basis. The optimal value is one. |

## 2.7 Predictive feature importance

We adopted the forward selection process from Löwe et al. (2021) to estimate the most important topographical predictive features for the U-Net model. Firstly, we trained 11 models, each of which considered one of the 11 topographical predictive features (precipitation depth was included in all models) from Table 2. Then, we evaluated the model performance based on
the performance indices in Table 3 and selected the best model. After that, we trained 10 new models based on the best model from the previous step by adding one of the remaining 10 predictive features to the inputs. We repeated this procedure three times to get the three most important predictive features for the U-Net model.

One of the advantages of the RF algorithm is the ability to compute the importance of predictive features, hence no forward selection process was required to estimate the importance of specific features for the RF models.

## 2.8 Computational details

The U-Net models were implemented using the Keras Python package (Chollet et al., 2015) while the RF models were implemented using the method ensemble.RandomForestRegressor from the Python package scikit-learn (Pedregosa et al., 2011). The U-Net models were trained using a high-performance machine with NVIDIA Quadro P4000 GPU while RF models were trained using a machine with Intel(R) Xeon(R) CPU E5-2667 v3@ 3.20 GHz. The training time ranged from 20 minutes to 48
180 hours and from 10 minutes to 2 hours for the U-Net and RF models respectively. The U-net models needed around 20 seconds to predict and map the water depth while the RF models took around 3 minutes.



## 3 Results and discussion

### 3.1 Evaluating different combinations of training data

In order to evaluate model transferability between spatial domains, we used a U-Net model with the following combination
of hyperparameters: depth = 4, kernel size = 3, number of filters in the first encoding block = 32. This combination showed
reasonable performance with the training datasets and had performed well in previous studies (Guo et al., 2021; Löwe et al.,
2021). For the RF model, we used the number of trees in the forest = 10 as it shows also reasonable results and training time
(around 10 minutes).

Figure 4 compares the performance indices for each study area (SA) and for all combinations of training/teesting datasets,
for both the U-Net and RF models. The NSE values show that the RF models outperformed the U-Net models for predicting
water depth within the training domains; however, they failed to predict water depth outside the training domains. It is obvious
from Figure 4 that the RF models were overfitted to the training data while the U-Net models tended to generalise better. The
CSI and RMSE values are in line with that finding: they show that the RF models could predict the inundation extent better
than the U-Net models in some training combinations despite failing to predict the water depth outside the training domain
accurately. Finally, it is clear from Figure 4 that the models U-Net - SA1 and RF - SA1 performed best outside the training
domain, compared to models trained using training data from the SA0 and SA2 separately. The U-Net-SA1 & 2 model had the
best performance within and outside the training domain.

### 3.2 Transfer learning

We evaluated how transfer learning could improve model performance outside the original training domain. Specifically, we
investigated how the improvement from transfer learning depends on the percentage of data that was used from the target
domain of the transfer. Figure 5 compares the performance of the transferred U-Net and RF models to the models trained
exclusively on the target domain of the transfer. The figure shows that the transfer learning technique boosted the U-Net and
RF model performance outside the training domain of the pre-trained models. Another advantage for transfer learning for
U-Net models is that training of the transferred models (20 minutes to two hours) was faster than training the whole network
from scratch.

All U-Net models transferred to the SA0 domain outperformed the U-Net-SA0 model for all performance indices. This
applies even if we used only 10 % of the available training data (from SA0) for transfer learning (in contrast to using 100
% of the SA0 training data for training the U-Net-SA0 model). We could conclude from Figure 5 that the transferred model
could use the previously learned knowledge from the U-Net-SA1&2 model to predict water depth in SA0. Contrary to U-Net,
the trained RF models for each SA separately outperformed all the transferred RF models. All RF models transferred to the
SA0 domain performed better than the RF-SA1&2 model, but worse than the RF-SA0 model. Figure 5 confirms the previous
findings that RF models are prone to overfitting.

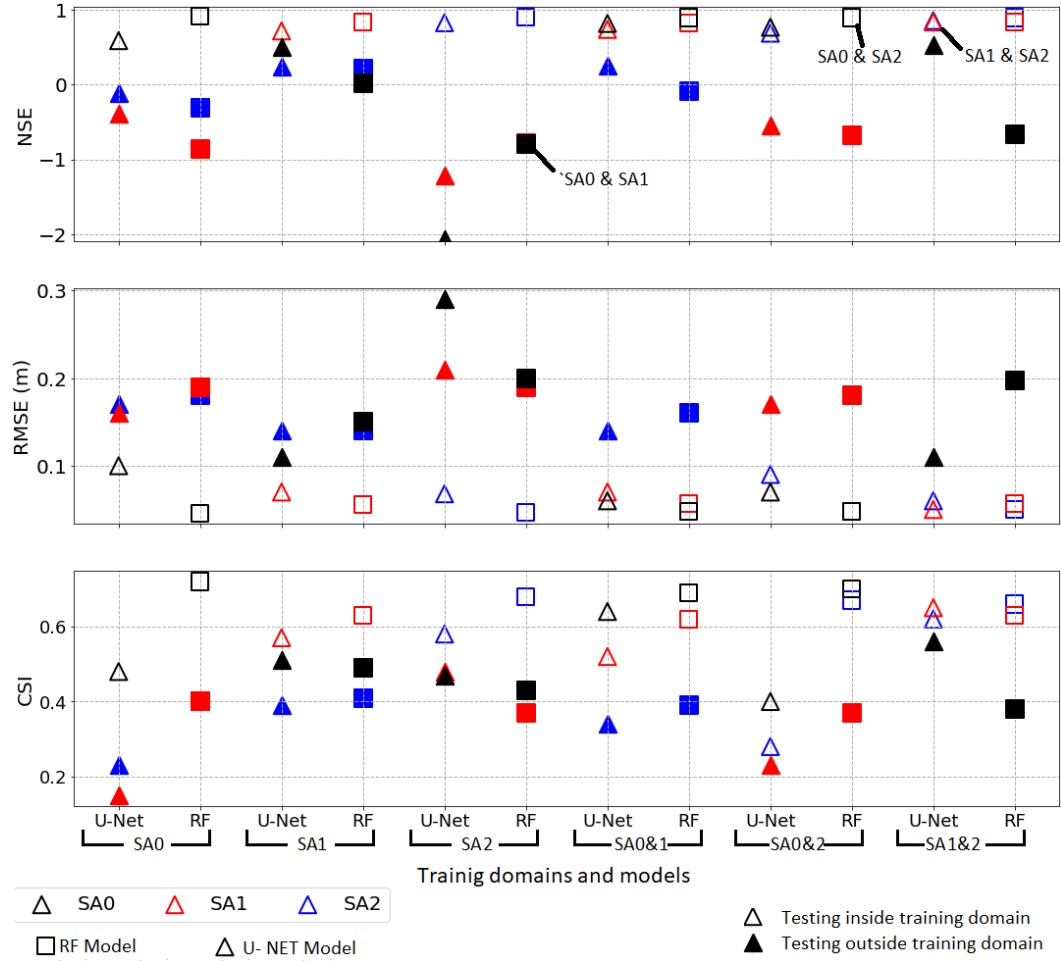

**Figure 4.** Computed performance indices (based on the testing dataset) for different combinations of training datasets for both the U-Net and RF models. The X-axis shows the used model and the training domain while the Y-axis shows the perfromacne indices. The U-Net-SA1 & 2 model had the best performance within and outside the training domain.

## 3.3 Flood maps

In order to convey a visual idea of the resulting flood maps, Fig. 6 compares the water depth as predicted by the different models to the water depths as simulated by the TELEMAC-2D model for region SA0 and for a precipitation depth of 100 mm. Apparently, all models could identify topographic depressions and predict flood water within them. The U-Net - SA0 model underestimates the water depth as shown in Figure 6b. Figure 6c and d show the predicted water depth from the best performance U-Net - SA1&2 model and the transferred model (U-Net - SA1&2 → SA0) using 10 % of the training data of SA0 (including only 40 and 120 mm precipitation depths) to train the weights in the transferred model. The transferred model outperformed both U-Net-SA0 and U-Net-SA1&2. It predicted the most identical inundation extent as the TELEMAC-2D model. Finally,




**Figure 5.** Evaluation of transfer learning: The colored markers represent the performance indices for transferred models with different percentages of data from the domain where the model has been transferred to. For example, SA0→SA1&2 refers to a model pre-trained on SA0, and then transferred (re-trained) on SA1 and SA2. The bars show the performance indices for the models trained exclusively on the transfer target domains. 10%* denotes that the training data from the transferred domain was generated using only two precipitation events (40 and 120 mm). The transferred U-Net-SA1&2→SA0 (pre-trained model SA1&2 and transfer target SA0) models outperformed the U-Net-SA0 model but the RF-SA0 model was superior to the transferred RF-SA1&2→SA0 models for all used percentages of new training data from SA0.

Figure 6e, f and g show the predicted water depth from the RF - SA0, RF - SA1&2 and RF - SA1&2 → SA0 models respectively. The RF - SA0 model memorised the training data as shown in Figure 4 and thus predicted the water depth accurately





while the RF - SA1&2 model could not predict the flood water inside the topographical depressions correctly and the RF - SA1&2 → SA0 model overestimated the water depth.

## 3.4  Feature importance

Figure 7 shows the NSE for SA1 and SA2 for the first three rounds in the predictive feature forward selection process for the best performance model U-Net-SA1&2 (other indices were computed but not shown here since the results regarding feature importance did not change). We stopped after three rounds because the process was computationally expensive and we aimed to obtain just the most important topographical predictive features. These were TWI, SDepth, roughness and altitude. TWI showed the best performance in the first round for both SA1 and SA2, while a model trained with TWI and SDepth was superior to other models in round two. Finally, training a model with TWI, SDepth and altitude outperformed the other models in round three. While the gained knowledge in round three by adding altitude and roughness was the same for SA1, adding roughness reduced the model performance in SA2. It is explainable that roughness influenced the models prediction because buildings were defined in the input dataset by having a high roughness values. The precipitation depth was added as a predictive feature to all the trained models but not included in Figure 7 because the main objective was to estimate the most important topographical predictive features. In contrast to (Löwe et al., 2021; Seleem et al., 2022), aspect was not among the most important features.

Figure 8 shows the feature importance for the RF-SA1&2 model. SDepth, altitude and CN were the most important predictive features. In contrast to U-Net models, TWI was not among the most important predictive features for the RF models. The estimated best predictive features from the U-Net and RF models were not the same but the results agree with the findings in the literature that TWI (Jalayer et al., 2014; Seleem et al., 2021; Bentivoglio et al., 2022),SDepth (Zhang and Pan, 2014; Seleem et al., 2021) and altitude (Zhang and Pan, 2014; Seleem et al., 2021, 2022) are indicators for urban flood-prone areas.

## 4  Conclusions

This study developed and tested CNN models (based on the U-Net architecture) and RF models to emulate the output of a 2-D hydrodynamic model (TELEMAC-2D) for three selected areas within the city of Berlin. We trained the data-driven surrogate models to map topographic, land cover and precipitation variables to flood water depths as obtained from 2D hydrodynamic model simulations. The evaluation of model performance was designed to assess the transferability of trained models to areas outside the training domain.

Both U-Net and RF models were skillful in predicting water depth within the training domain (minimum NSE=0.6). Contrary to the hypothesis that deep learning algorithms were superior to shallow machine learning algorithms (Bentivoglio et al., 2022), the results suggested that the RF models outperformed the U-Net models for predictions within the training domain. However, we found that the high performance of RF models was largely owed to overfitting: outside of the training domains, RF models exhibited a substantial performance loss for all considered metrics (NSE, RMSE, and CSI). For the CNN models, the a loss of



**Figure 6.** Comparison of water depths from different models and TELEMAC-2D model for a 100 mm precipitation event for SA0. The figure highlights the boundary of two topographic depressions within SA0 where runoff accumulates. The altitude is shown in the background.





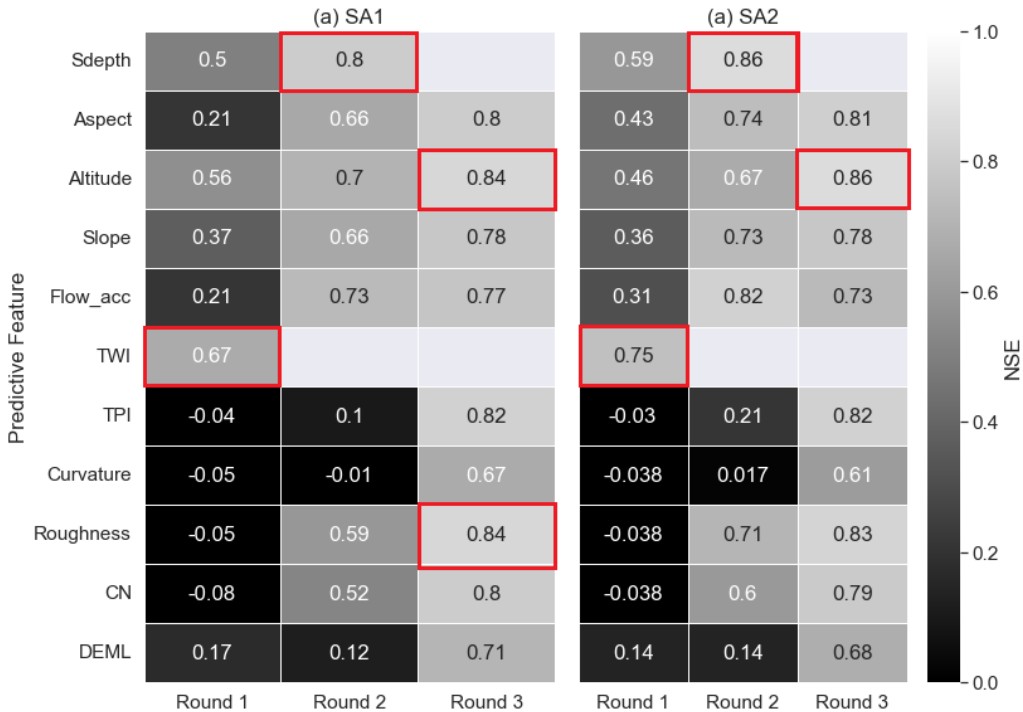

**Figure 7.** NSE values for SA1 (a) and SA2 (b) for the models trained in the forward selection process for the best performance training data combination (U-Net - SA1&2). The best performance model in every round is marked in red.

performance was also considerable, but clearly less pronounced than for the RF models. We hence conclude that the potential

of CNN models to generalize beyond the training domain is significantly higher than for RF models.

Furthermore, we found that the CNN models' ability to generalize and hence to be transferred beyond the training domain could be boosted by transfer learning: by providing only a small fraction of training data from a target domain, transfer learning improved the performance of some a pre-trained CNN models in a way it outperformed a CNN that was trained from scratch with the full amount of training data from that domain. This outcome clearly distinguishes deep learning models such as CNN

from shallow models such as RF which could not benefit from transfer learning in a similar fashion. Transfer learning hence provides a promising perspective to efficiently use additional training data to adjust deep learning models to specific target areas or to provide an additional level of generalization, at a minimum of computational expense.

Analyzing the results showed that the depth of a depression (SDepth) is a strong predictive feature for both the U-Net and RF models. SDepth, altitude and CN were the most influencing topographical predictors for the RF model while TWI,

SDepth,roughness and altitude were the most influencing topographical predictive features for the U-Net model. This is in contrast to Löwe et al. (2021) and Seleem et al. (2022) who found the aspect to be the most important predictive feature for

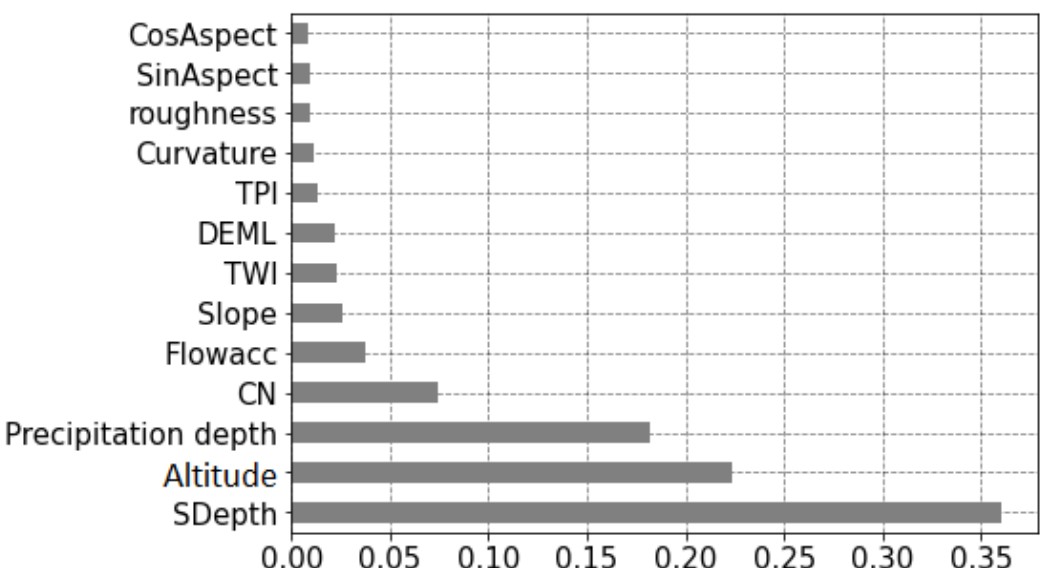

**Figure 8.** Predictive feature importance for RF-SA1&2 model.

flood hazard and susceptibility mapping using CNN. We hence suggest a detailed future study to systematically explore the suitability of different topographical predictive features for data-driven models of urban flood hazard.

Altogether, this study confirms that deep learning could be a skilful tool for upscaling flood hazard maps in urban envi-
ronments. Given the excessive costs of providing complete high-resolution 2-D hydrodynamic model coverage, deep learning, namely CNN, has shown the ability to learn transferable knowledge of simulated inundation patterns. This puts into perspective previous study results by Seleem et al. (2022) that highlighted the performance of random forest models – which we now found less able to generalize. Given the apparent potential of CNN for generalization, however, it is all the more important to collect training and testing data from many and diverse regions in order to capitalize on this learning capability. This could be
a community effort, and the basis for future benchmarking experiments that move beyond the boundaries of isolated cities. In order to start this process, we provided the output of the 2-D hydrodynamic simulations along with this paper.

## Appendix A:  Hyperparameter tuning

Figure A1,A2, and A3 show the computed performance indices for different combinations of U-Net hyper-parameter (depth, number of filters in the first encoder block, and filter size) combinations using the best performance training data combination
(SA1&2). We considered a total of 18 hyper-parameter combinations and selected the best combination based on the model performance in the training domain. Figure A2, and A3 shows that the best performance model for both SA1 and SA2 (study areas included in the training domain) had a number of filters =32, filter size = 3, and depth = 4. It has 7,771,809 parameters and a training time of 8 hours 21 minutes.



We trained RF models with different numbers of trees in the forest also using the best performance training data combination
(SA1&2), increasing the number of trees in the forest increased the computational cost and the training time (from 10 minutes
to 3 hours) but it had no significant performance gain on the model prediction. Our findings agree with previous studies (Oshiro
et al., 2012; Zahura et al., 2020). Therefore, we used the number of trees in the forest = 10 for further predictions in this paper.

*Code and data availability.* The predictive features, water depth from the TELEMAC 2D model simulations are available at https://doi.
org/10.5281/zenodo.7221058 (last access: 18 October 2022); the source code for the models are provided through a GitHub repository
https://github.com/omarseleem92/Urban_flooding.git.

*Author contributions.* OS, GA and MH conceptualized this study. OS developed the software and carried out the formal analysis; OS and
MH prepared the manuscript with contribution from GA and AB.

*Competing interests.* The authors declare that no competing interests are present

*Acknowledgements.* This research was funded by the Deutscher Akademischer Austauschdienst (DAAD).We acknowledge the support of
295 the Deutsche Forschungsgemeinschaft and Open Access Publishing Fund of the University of Potsdam.





| | Filter_num | Filter_size | Depth | NSE | RMSE | CSI |
|---|---|---|---|---|---|---|
| 0 | 16 | 3 | 3 | 0.216079 | 0.144989 | 0.534608 |
| 1 | 16 | 3 | 4 | 0.450139 | 0.121430 | 0.506848 |
| 2 | 16 | 5 | 3 | 0.405210 | 0.126293 | 0.528408 |
| 3 | 16 | 5 | 4 | 0.465831 | 0.119684 | 0.539143 |
| 4 | 16 | 7 | 3 | 0.602343 | 0.103265 | 0.542547 |
| 5 | 16 | 7 | 4 | 0.429672 | 0.123669 | 0.491252 |
| 6 | 32 | 3 | 3 | 0.079667 | 0.157098 | 0.518367 |
| 7 | 32 | 3 | 4 | 0.531396 | 0.112099 | 0.555347 |
| 8 | 32 | 5 | 3 | 0.533752 | 0.111817 | 0.538400 |
| 9 | 32 | 5 | 4 | 0.431954 | 0.123421 | 0.512028 |
| 10 | 32 | 7 | 3 | 0.141585 | 0.151721 | 0.527111 |
| 11 | 32 | 7 | 4 | -0.068188 | 0.169247 | 0.503513 |
| 12 | 64 | 3 | 3 | 0.530566 | 0.112198 | 0.511400 |
| 13 | 64 | 3 | 4 | 0.349439 | 0.132082 | 0.536829 |
| 14 | 64 | 5 | 3 | 0.415645 | 0.125180 | 0.525844 |
| 15 | 64 | 5 | 4 | 0.387801 | 0.128128 | 0.531205 |
| 16 | 64 | 7 | 3 | 0.450000 | 0.121345 | 0.510000 |
| 17 | 64 | 7 | 4 | 0.470000 | 0.121345 | 0.500000 |

**Figure A1.** Calculated performance indices for SA0 for all the computed hyperparameter combinations for the best training dataset combination (SA1 & 2).



| | Filter_num | Filter_size | Depth | NSE | RMSE | CSI |
|---|---|---|---|---|---|---|
| 0 | 16 | 3 | 3 | 0.738003 | 0.070887 | 0.593455 |
| 1 | 16 | 3 | 4 | 0.715503 | 0.073868 | 0.537106 |
| 2 | 16 | 5 | 3 | 0.765911 | 0.067005 | 0.575877 |
| 3 | 16 | 5 | 4 | 0.724726 | 0.072661 | 0.575724 |
| 4 | 16 | 7 | 3 | 0.633788 | 0.083808 | 0.459598 |
| 5 | 16 | 7 | 4 | 0.772377 | 0.066073 | 0.561517 |
| 6 | 32 | 3 | 3 | 0.794891 | 0.062720 | 0.618499 |
| 7 | 32 | 3 | 4 | 0.837990 | 0.055743 | 0.651536 |
| 8 | 32 | 5 | 3 | 0.729301 | 0.072054 | 0.527954 |
| 9 | 32 | 5 | 4 | 0.703284 | 0.075438 | 0.479789 |
| 10 | 32 | 7 | 3 | 0.810939 | 0.060217 | 0.595246 |
| 11 | 32 | 7 | 4 | 0.779307 | 0.065060 | 0.590154 |
| 12 | 64 | 3 | 3 | 0.695888 | 0.076372 | 0.548835 |
| 13 | 64 | 3 | 4 | 0.760771 | 0.067737 | 0.604587 |
| 14 | 64 | 5 | 3 | 0.759953 | 0.067853 | 0.587598 |
| 15 | 64 | 5 | 4 | 0.833772 | 0.056464 | 0.605823 |
| 16 | 64 | 7 | 3 | 0.669744 | 0.079587 | 0.515753 |
| 17 | 64 | 7 | 4 | 0.740098 | 0.070604 | 0.558408 |

**Figure A2.** Calculated performance indices for SA1 for all the computed hyperparameter combinations for the best training dataset combination (SA1 & 2)..



| | Filter_num | Filter_size | Depth | NSE | RMSE | CSI |
|---|---|---|---|---|---|---|
| 0 | 16 | 3 | 3 | 0.728378 | 0.086787 | 0.551695 |
| 1 | 16 | 3 | 4 | 0.683640 | 0.093662 | 0.491272 |
| 2 | 16 | 5 | 3 | 0.775044 | 0.078981 | 0.551749 |
| 3 | 16 | 5 | 4 | 0.745083 | 0.084076 | 0.537122 |
| 4 | 16 | 7 | 3 | 0.565798 | 0.109728 | 0.377808 |
| 5 | 16 | 7 | 4 | 0.795849 | 0.075240 | 0.548188 |
| 6 | 32 | 3 | 3 | 0.853574 | 0.063721 | 0.611738 |
| 7 | 32 | 3 | 4 | 0.859656 | 0.062384 | 0.620996 |
| 8 | 32 | 5 | 3 | 0.737983 | 0.085239 | 0.492455 |
| 9 | 32 | 5 | 4 | 0.747101 | 0.083743 | 0.462011 |
| 10 | 32 | 7 | 3 | 0.841518 | 0.066292 | 0.592339 |
| 11 | 32 | 7 | 4 | 0.810015 | 0.072583 | 0.594880 |
| 12 | 64 | 3 | 3 | 0.665841 | 0.096267 | 0.479617 |
| 13 | 64 | 3 | 4 | 0.767345 | 0.080326 | 0.579144 |
| 14 | 64 | 5 | 3 | 0.770667 | 0.079751 | 0.542615 |
| 15 | 64 | 5 | 4 | 0.856114 | 0.063170 | 0.584332 |
| 16 | 64 | 7 | 3 | 0.685174 | 0.093441 | 0.489645 |
| 17 | 64 | 7 | 4 | 0.729622 | 0.086594 | 0.474623 |

**Figure A3.** Calculated performance indices for SA2 for all the computed hyperparameter combinations for the best training dataset combination (SA1 & 2)..



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
