# Peer review of "Transferability of data-driven models to predict urban pluvial flood water depth in Berlin, Germany"

_Natural Hazards and Earth System Sciences, 2022_

## Referee Comment (RC2)

**Review Comments**

The author attempts to use RF, CNN and migration learning methods to solve the problem of expanding the application of data-driven models. This problem is very valuable for urban flooding research, and is also the direction of machine learning and physical model development. Taking three typical regions in Berlin, Germany as the research object, the author constructs a waterlogging depth prediction model using CNN and RF. The results show that the CNN models had significantly higher potential than the RF models to generalize beyond the training domain. But there are still some problems to be solved.

1. Table 2: I have not found the indicators related to the drainage pipe network in the in the table. The urban flood not only depends on the rainfall, terrain, slope and elevation, but also the drainage capacity of the drainage pipe network is an important factor. Please consider whether it is better to add the relevant indicators of the drainage pipe network.

2. Why to select a rainfall event with a rainfall duration of 1 hour? It is suggested to give a reasonable explanation on the basis of statistical regional rainfall characteristics.

3. As far as I know, recent floods will be easier to collect and more valuable. Why is the flood inventory only collected in 2017? If so, suggestions can be collected and supplemented.

4. The data of hydrodynamic simulation was selected as the sample data of the data-driven model to establish the water depth prediction model. Therefore, if the reliability of the hydrodynamic simulation results can be verified, the data-driven model established will have high value for urban flood control.

5. In Figure 7 (a), it can see that the altitude and roughness have the same value. If the decimal places are more accurate, can this be avoided? In addition, the author mentioned that adding roughness will reduce the performance in SA2. Therefore, whether it is possible to delete roughness directly?

6. The analysis in Section 3.3 is an intuitive feeling. It is suggested to add some statistical indicators to make it more convincing.

7. Although Berlin is large, the research area selected by the author seems to be small, and the model training time is from 20 minutes to 48 hours. If a large city (more than 1000km2) is modeled, the calculation cost seems too high.

8. Transfer learning should be the focus of this study, but the analysis of transfer learning methods and results in 2.5 and 3.2 is insufficient and needs to be supplemented.

---

## Author Comment (AC1)

Dear referees, dear editor,

We would like to thank you for your comments and constructive suggestions to our manuscript. We very much appreciate the time and effort that you have invested in your reports. This letter contains the responses to the referee comments which also formed the basis for the revision of the manuscript. We sincerely hope that the revised version is now acceptable for publication in Natural Hazards and Earth System Sciences, and we are looking forward to a decision.

Kind regards,

Omar Seleem

(on behalf of the author team)

**Response to referee #1**

**RC: This paper is concerned with the comparison of two machine-learning techniques for predicting urban pluvial flood hazard in areas where the machine learning model was not trained. Namely, these are the UNET approach that has recently been used in a number of studies for the same purpose, and the random forest approach that was similarly used for predicting flooding, but (to my knowledge) not in a context of generating 2D flood maps for specific rain events. In addition, the papers explores the probability of transfer learning for these approaches**

**The paper is interesting, generally well written and within the scope of the journal. I do have one major criticism and a number of minor comments as detailed below, but I expect that these can be addressed in a revision and that the paper can subsequently be recommended for publication.**

AR: We thank the referee for the positive feedback, and the recommendations.

**RC: Major comments:**

**The comparison of UNET and random forest (RF) is a (the) key aspect of the paper. The paper concludes that the RF does not transfer well to other areas because it overfits the training data. This effect can and should have been avoided, and since it is such a central part of the paper I do think it needs to be fixed.**

**The straightforward approach would be to apply a cross validation approach. Divide the training dataset into, for example, 5 areas, perform 4 training iterations where you train on 4 areas and validate on the 5th, select RF hyperparameters that minimize the cross validation loss, and then train with these hyperparameters on the entire dataset.**

AR: We thank the referee for these recommendations. We used holdout validations instead of k-fold cross validation. The latter is expected to avoid misleading results. However, it is very computationally

expensive to use k-fold cross validation for all implemented convolutional neural network (CNNs) and random forest (RF) models. In addition, Löwe et al., 2021 showed that a model trained using the holdout validations method was superior to models trained using the k-fold cross validation to predict urban floodwater depth using CNNs. K-fold cross validation method increased the prediction error because of insufficient representation of the whole dataset. For the RF models, we used a relatively large training dataset (10 % of the available training data (number of pixels within the training domain × number of training precipitation events)). Therefore, we considered it acceptable to use the holdout validations as we trained several models and used relatively big training datasets which helped to avoid outliers (Löwe et al., 2021; Guo et al., 2022).

To address the referee comment, we carried out a hyper-parameter tuning (number of iterations = 30) for the best performing model (RF − SA1&2) with k-fold cross-validation and a smaller training dataset (number of samples =100,000) as shown in Table 1. Then, we trained a model with the best hyper-parameters combination on the entire dataset. The trained model performance within the training domain reduced slightly (Nash Sutcliffe Efficiency (NSE) values dropped from 0.84 and 0.9 to 0.75 and 0.84 for SA1 and SA2 respectively). Outside the training domain (the NSE increased from -0.67 to 0.03 for SA0). The calculated performance indices showed that the model trained using the best hyperparameter combination and k-fold cross-validation still could not generalize outside the training domain. We suggest adding the table to the supplementary file and mention it in the revised manuscript if the referee agrees.

**In particular, this concerns the depth of the decision trees. As I understand from the paper, these have not been limited, and therefore the trees probably simply incorporate the entire training dataset. Please do include results for this in the paper or appendix.**

AR: We agree with the referee that not limiting the depth of the decision trees may cause overfitting. We investigated the impact of limiting the depth of decision trees on the holdout validation method. We implemented several models (RF- SA1&2) with varying the depth of the decision tree. Figure 1 shows the computed NSE values for all the implemented models. It points out that reducing the maximum depth of the decision tree enhanced the model performance outside the training domain at the cost of reducing the model performance inside the training domain.

The goal of our study is to assess the model transferability in space. To do this, we selected the best combination of hyperparameters for each model based on their performance on the validation dataset. We then evaluated the models' performance on the testing dataset. We did not specifically choose the hyperparameter combination to optimize the models' transferability to other regions, Figure A1 in the appendix shows that a CNN model with 16 filters, a filter size of 7, and a network depth of 3 was superior for predictions outside of the training domain but had lower performance within the training domains. Based on the aforementioned analysis, we recommend including this figure in the supplement and mentioning it in the revised manuscript. However, we also suggest keeping the results we obtained from the RF models, if the referee agrees.

[Figure]

Figure 1 shows the impact of varying the maximum depth of the decision trees on the RF – SA1 &2 model performance using holdout validation method. The X-axis shows the implemented maximum depth of the decision trees while the Y-axis denotes the computed Nash Sutcliffe Efficiency (NSE).

Table 1. Selection of best combination of parameters for the RF model based on hyperparameter tuning using K-fold cross-validation method.

| Parameter | Range | Best combination |
|---|---|---|
| Number of trees in random forest | 100 < n_estimator< 2000 | 2000 |
| Number of features to consider at every split | [ auto , sqrt] | auto |
| Maximum number of levels in tree | 10 < max_depth < ∞ | 50 |
| Minimum number of samples required to split a node | min_samples_split = [2, 5, 10] | 5 |
| Minimum number of samples required at each leaf node | min_samples_leaf = [1, 2, 4] | 2 |
| Method of selecting samples for training each tree | bootstrap = [True, False] | True |

**RC: Minor comments:**

**Figure 1: The different subareas have quite different properties (models trained on some areas generalize, while this is not the case for other areas). Please include som detailed illustrations of the**

**subareas (elevation, maybe also flood areas). I think this Figure could become a 4 panel figure, one panel showing the overview of Berlin and the other panels details of the 3 areas. The legend needs to include units and an explanation that it is elevation that is displayed**.

AR: We modified the figure in the revised manuscript by dividing the figure into four panels and showed the water depth map from TELEMAC-2D simulations for a 100 mm effective rainfall event for the three study areas with the elevation map in the background.

**RC: line 92: what kind of storms were fed into the hydrodynamic simulations? block rains? CDS? Euler? ...**

AR: We used one-hour block rainfall with precipitation depth ranging from 20 to 150 mm (10 mm steps) effective rainfall to perform the two-dimensional hydrodynamic model.

**RC: line 173: please provide details on how feature importance was assessed (remove variables and assess relative change of loss function? validation loss or training loss?)**

AR: We used the built-in feature importance in random forest, which is implemented in scikit-learn python module (Pedregosa et al., 2011). The feature importance are calculated as the mean and standard deviation of accumulation of the impurity decrease within each tree (Pedregosa et al., 2011).

**RC: Figure 4: This figure is impossible to read in black and white print**

**While all the information in the figure is highly relevant, it is also quite convoluted and hard to understand. My guess is that it will be easier to read if you group all UNET results on the left, and all RF results on the right. "UNET" and "RF" could then be moved as headings to the top of the figure, making it a bit easier to recognize the "training domains" text as an axis caption (there is also a typo in the figure). You can then also use the same symbol for RF and UNET results, making the legend consistent with the symbols in the figure.**

AR: We thank the referee for his recommendation. We agree with the referee that it is hard to read the figure in black and white. Actually, we had implemented the referee's recommendation in earlier draft versions as shown in Figure 2. However, we found that joining the results from U-Net and RF models together in one panel allows for a more direct comparison between the models. If the referee agrees, we suggest keeping the original figure (in colors) in the revised manuscript.

[Figure]

Figure 2. Computed performance indices (based on the testing dataset) for different combinations of training datasets for both the U-Net and RF models. The SA1 & 2 Model had the best performance within and outside the training domain.

**RC: Figure 5:**

**I would suggest rearranging this figure in the same way as Fig. 4. Consider also including some more text into the figure, e.g. "baseline training areas" and "transfer test area" so that the figure becomes a bit more self-sufficient. Symbols like SA0->SA1&2 make it very hard to understand the figure without reading everything else in detail**.

AR: We thank the referee for his suggestion. We agree with the referee that the figure requires attention to be understood. Therefore, we used the figure caption to clarify and explain the figure. The referee's suggestion is applicable for Figure 4 (in the manuscript) as shown in Figure 2 (in the response letter). However, it is difficult to apply it for Figure 5 (in the manuscript). Some columns would have ten values (for example, SA0 column would have four points representing the different amount of data used in the transferred model and one point representing the model trained exclusively on the transfer target domains (represented as bars in the original manuscript) for SA1 and similarly for SA2). Hence, it will be difficult to combine the results from U-Net and RF models together as the referee suggested. Therefore, we regret to suggest again keeping the original version of the figure.

**RC: Figure 6:**

**Please include a similar figure for a smaller event. The most challenging part for machine learning models is to correctly capture the boundary between flood / no flood. In general, I miss results for how well the models perform for different rain intensities**.

AR: We thank the referee for his suggestion. The model performance varies with the inundation extent. The model performance enhances with increasing the inundation area. We suggest adding similar figures for the 50 and 140 mm effective rainfall events in the supplementary and refer to them in the revised manuscript if the referee agrees.

**Response to referee #2**

**RC: The author attempts to use RF, CNN and machine learning methods to solve the problem of expanding the application of data-driven models. This problem is very valuable for urban flooding research, and is also the direction of machine learning and physical model development. Taking three typical regions in Berlin, Germany as the research object, the author constructs a waterlogging depth prediction model using CNN and RF. The results show that the CNN models had significantly higher potential than the RF models to generalize beyond the training domain. But there are still some problems to be solved.**

AR: We thank the referee for the feedback and the recommendations. The main aim of this study is to investigate the transferability of data-driven models in space to predict urban pluvial floodwater depth. Hydrodynamic models are used to generate data to implement data-driven models due to the absence of measurement gauges in urban areas (Bentivoglio et al., 2022). The two-dimensional hydrodynamic models are considered the best representation of the physical process of runoff generation and concentration and are commonly used in the literature without validation (Löwe et al., 2021; Guo et al., 2022; Bentivoglio et al., 2022).

**1. Table 2: I have not found the indicators related to the drainage pipe network in the in the table. The urban flood not only depends on the rainfall, terrain, slope and elevation, but also the drainage capacity of the drainage pipe network is an important factor. Please consider whether it is better to add the relevant indicators of the drainage pipe network.**

AR: We thank the referee for his recommendation. Urban pluvial flooding is caused by short intensive precipitation events that exceed the capacity of the urban storm drainage system (Houston et al., 2011). Storm drainage systems in many cities are old, need maintenance, and the inlet gullies are often blocked (Fathy et al., 2020). Their detailed drawings are often not available (Rangari et al., 2018; Hou et al., 2021). Furthermore, the city of Berlin has a relatively flat topography and Van Dijk et al., 2014 showed that there was no difference in the results of 2D and coupled 1D-2D hydrodynamic models in urban flat areas. Considering the aforementioned reasons, the unavailability of detailed information about the storm drainage system in Berlin, and previous studies (Hou et al., 2021; Löwe et al., 2021; Guo et al., 2022), we neglected a detailed consideration of the drainage system effects on the TELEMAC-2D simulations. Instead, as a first-order approximation, we considered the retention effects of the drainage system and other water retention processes in the urban landscape to be dealt with the effective rainfall approach, which we followed in this paper. I.e only the rainfall share, which converts to urban overland flow, is considered here.

This study developed data-driven models to surrogate the two-dimensional hydrodynamic models. While data-driven models do not "understand" the physical processes of runoff generation and concentration, they are designed to detect relationships between input and target variables (Grant and Wischik, 2020). Therefore, we used 12 predictive features based on the literature (Löwe et al., 2021; Guo et al., 2022; Bentivoglio et al., 2022). We agree with the referee that including storm drainage system in both the hydrodynamic simulations and as a predictive feature to develop the data-driven model would better represent the reality. However, as detailed data about the storm drainage system for the study area were not available, we follow the aforementioned effective rainfall approach.

**2. Why to select a rainfall event with a rainfall duration of 1 hour? It is suggested to give a reasonable explanation on the basis of statistical regional rainfall characteristics.**

AR: We thank the referee for his suggestion. Urban pluvial flooding is typically caused by convective rainfall events that occur on small spatial and temporal scales. They could occur anywhere even in areas without flooding history. In many cases, the rain gauges are located far from the rainfall and hence they do not capture the maximum precipitation depth as in Berlin 2019 (Berghäuser et al., 2021) and southern Germany 2016 (Bronstert et al.,2017; Bronstert et al.,2018) flood events. We used one-hour effective rainfall events to perform the TELEMAC-2D simulations for the following reasons:

1- Some rain gauges measured more than 30 mm rainfall depth within one hour in Berlin in 2019 flood event and the rainfall radar analysis showed that the gauges were located far away from the rainfall location (Berghäuser et al., 2021) which means that the actual rainfall depth was higher.
2- We used a 1 m grid mesh to perform the TELEMAC-2D simulation to represent the complex urban environment. This fine grid increased the computational time of the hydrodynamic simulation. Using a longer rainfall duration will increase the computational time.

Considering that the main aim of this study is to investigate the transferability of data-driven model in space to predict urban pluvial flood and the aforementioned reasons, we believe that statistical regional rainfall analysis is out of the scope of this study.

**3. As far as I know, recent floods will be easier to collect and more valuable. Why is the flood inventory only collected in 2017? If so, suggestions can be collected and supplemented.**

AR: We thank the referee for his suggestion. Berlin Wasserbetriebe (BWB) is the authority that is responsible for managing storm drainage in Berlin. They collect the flood inventory for the city of Berlin based on reports of the fire brigade, from social media and customer reports between the years 2005 to 2017 (Seleem et al., 2022). Recent floods are being documented however, they are still not available. While the flood inventory was useful to identify the study areas, having an up-to-date flood inventory is not influencing the main aim of this study because we generated the datasets based on the TELEMAC-2D model simulations and the selected study areas are known flood-prone areas.

**4. The data of hydrodynamic simulation was selected as the sample data of the data-driven model to establish the water depth prediction model. Therefore, if the reliability of the hydrodynamic simulation results can be verified, the data-driven model established will have high value for urban flood control**

AR: We agree with the referee that validation of the hydrodynamic simulations would make the simulations more reliable. However, unfortunately, no measurements, aerial photographs or satellite data of pluvial flooding events in the past are available and, hence, we could not validate the hydrodynamic model results. As mentioned above, the two-dimensional hydrodynamic models are considered the best representation of the physical process of runoff generation and concentration and are commonly used in the literature without validation (Löwe et al., 2021; Guo et al., 2022; Bentivoglio et al., 2022).

**5. In Figure 7 (a), it can see that the altitude and roughness have the same value. If the decimal places are more accurate, can this be avoided? In addition, the author mentioned that adding roughness will reduce the performance in SA2. Therefore, whether it is possible to delete roughness directly?**

AR: We thank the referee for his suggestion. We checked for two decimals as shown in Table 2. Table 2 shows that while adding altitude is slightly improving the model prediction than roughness based on RMSE and CSI, adding roughness to the predictive features was better based on NSE values. Furthermore, as mentioned in the manuscript we defined the buildings by giving them a high roughness values. Therefore, we believe it is important to show that the predictive feature that defines the buildings is influencing the model prediction and hence highlight the importance of well representing the buildings for future research. Thus, we suggest keeping the roughness as an important predictive feature if the referee agrees.

Table 2 Calculated Performance indices from the forward selection round 2 for SA1.

|          | Altitude | Roughness |
|----------|----------|-----------|
| NSE      | 0.838    | 0.84      |
| RMSE (m) | 0.055    | 0.056     |
| CSI      | 0.650    | 0.646     |

**6. The analysis in Section 3.3 is an intuitive feeling. It is suggested to add some statistical indicators to make it more convincing.**

AR: We agree that section 3.3 is more of a qualitative assessment. The motivation is to provide a visual impression of how the different models capture inundation patterns (here based on the TELEMAC-2D simulation of a 100 mm effective precipitation event) in the three different study areas. In our opinion, such a visual assessment is important to appraise the plausibility of model behavior. The formal statistical evaluation, however, is part of the verification analysis presented in sections 3.1 and 3.2.

**7. Although Berlin is large, the research area selected by the author seems to be small, and the model training time is from 20 minutes to 48 hours. If a large city (more than 1000km2) is modeled, the calculation cost seems too high.**

AR: We thank the referee for this comment. We agree with the referee that deep learning models have high computational cost and the model accuracy is influenced by finding the best hyperparameter combination that fits the problem and the available dataset. The model's training time depends on several factors (e.g., the model architecture, number of filters, filter size, number of trainable parameters, training dataset size, etc.). Furthermore, Löwe et al., 2021 trained a convolutional neural network model to predict water depth for the city of Odense (Denmark). The city area is around 305 $km^2$ and the training time ranged from 6.5 hours to 5 days. However, once a model is trained, it predicts floodwater depth within seconds (minutes for a whole city) as mentioned in Line 180 in the original manuscript. Hence a trained model is useful for carrying out not only a city scale's flood risk management but also operational management.

**8. Transfer learning should be the focus of this study, but the analysis of transfer learning methods and results in 2.5 and 3.2 is insufficient and needs to be supplemented.**

AR: We thank the referee for his comment. Actually, the main aim of this study is investigating the transferability of data-driven models outside the training domain. We compared the performance of deep learning with random forest because random forest showed superiority to map urban flood susceptibility in previous study (Seleem et al., 2022). Transfer learning is a tool that we used to enhance the model transferability to new areas and to show that the trained deep learning models could be adapted to new data-scare areas benefiting from the knowledge learned from the pre-trained model. We suggest making the aim of the study clear in the last paragraph in the introduction if the referee agrees.

**References:**

Bentivoglio, R., Isufi, E., Jonkman, S. N., and Taormina, R.: Deep Learning Methods for Flood Mapping: A Review of Existing Applications and Future Research Directions, Hydrology and Earth System Sciences Discussions, pp. 1–50, 2022

Berghäuser, L., Schoppa, L., Ulrich, J., Dillenardt, L., Jurado, O. E., Passow, C., ... & Thieken, A. H. (2021). Starkregen in Berlin: Meteorologische Ereignisrekonstruktion und Betroffenenbefragung.

Bronstert, A., Agarwal, A., Boessenkool, B., Fischer, M., Heistermann, M., Köhn-Reich, L., Moran, T., Wendi, D. (2017): Die Sturzflut von Braunsbach am 29. Mai 2016 - Entstehung, Ablauf und Schäden eines "Jahrhundertereignisses". Teil 1: Meteorologische und Hydrologische Analysen. Hydrologie und Wasserbewirtschaftung, 61(3), 150-162.

Bronstert, A., Agarwal, A., Boessenkool, B., Crisologo, I., Fischer, M., Heistermann, M., Köhn-Reich, L., López-Tarazón, J.A., Moran, T., Ozturk, U., Reinhardt-Imjela, C., Wendi, D. (2018): Forensic hydro-meteorological analysis of an extreme flash flood: The 2016-05-29 event in Braunsbach, SW Germany. Science of the Total Environment, 630, 977-991. doi.org/10.1016/j.scitotenv.2018.02.241

Fathy, I., Abdel-Aal, G. M., Fahmy, M. R., Fathy, A., & Zeleňáková, M. (2020). The negative impact of blockage on storm water drainage network. Water, 12(7), 1974.

Guo, Z., Moosavi, V., and Leitão, J. P.: Data-driven rapid flood prediction mapping with catchment generalizability, Journal of Hydrology, 609, 127 726, 2022.

Hou, J., Yang, D., Li, B., Bai, G., Xia, J., Wang, Z., & Su, F. (2021). Approximate Method for Evaluating the Drainage Process of an Urban Pipe Network with Unavailable Data. Journal of Irrigation and Drainage Engineering, 147(10), 04021043.

Houston, D., Werrity, A., Bassett, D., Geddes, A., Hoolachan, A. & McMillan, M. (2011), 'Pluvial (rain-related) flooding in urban areas: the invisible hazard'.

Löwe, R., Böhm, J., Jensen, D. G., Leandro, J., and Rasmussen, S. H.: U-FLOOD–topographic deep learning for predicting urban pluvial flood water depth, Journal of Hydrology, 603, 126 898, 2021.

Rangari, V. A., Veerendra Gopi, K., Umamahesh, N. V., & Patel, A. K. (2018). Simulation of urban drainage system using disaggregated rainfall data. In Hydrologic modeling (pp. 123-133). Springer, Singapore.

Van Dijk, E., van der Meulen, J., Kluck, J., & Straatman, J. H. M. (2014). Comparing modelling techniques for analysing urban pluvial flooding. Water science and technology, 69(2), 305-311.

Seleem, O., Ayzel, G., de Souza, A. C. T., Bronstert, A., and Heistermann, M.: Towards urban flood susceptibility mapping using data-driven models in Berlin, Germany, Geomatics, Natural Hazards and Risk, 13, 1640–1662, 2022.

---

## Author Response (AR2)

Dear referees, dear editor,

We would like to thank you for your comments and constructive suggestions to our manuscript. We very much appreciate the time and effort that you have invested in your reports. This letter contains the responses to the referee comments which also formed the basis for the revision of the manuscript. We sincerely hope that the revised version is now acceptable for publication in Natural Hazards and Earth System Sciences, and we are looking forward to a decision.

Kind regards,

Omar Seleem

(on behalf of the author team)

**Response to referee #1**

**The authors have addressed my comments. I suggest accepting the paper with some suggestions for minor changes:**

AR: We thank the referee for the positive feedback, and the recommendations.

**Fig. 7 - it is not clear to me why feature importance is assessed on the training data. These results may not provide us any insight into which variables are relevant for generating predictions in other areas. Would it be relevant to include the NSE for SA0 in the figure?**

AR: We assessed the importance of the predictive features based on the testing dataset inside the training domains (precipitation depths which were not included in the training dataset) because the importance of the predictive features for predictions in other areas outside the training domain varies with the characteristics of these areas. For example, Figure 1 shows that while the topographic wetness index (TWI) was the most influencing predictive feature in the first round for SA1 and SA2, the depth of topographic depressions (SDepth) was the most influencing predictive feature for SA0. Therefore, we would like to suggest keeping Figure 7 (in the manuscript) without modifications if the referee agrees.

[Figure]

Fig1. NSE values for SA0 (a), SA1 (b) and SA2 (c) for the models trained in the forward selection process

for the best performance training data combination (U-Net - SA1&2). The best performance model in every round is marked in red.

**Conclusions - I think one of the main conclusions from Fig. 4 is that an appropriate composition of the training dataset is very important for the performance of the model. This is a general principle with data-driven models where the training data need to cover the range of situations for which we want to predict (e.g. a model trained only in Berlin will not work in the Alps). I suggest adding a sentence on this.**

AR: We agree with the referee. Applying a data-driven model for flood prediction is still a rising topic. The literature still lacks such a study which investigates how a data-driven model trained for a relatively topographically flat city as Berlin, performs in cities with different characteristics (cities in mountainous areas). We add the following sentence to the conclusion "Further research requires testing the data-driven model's transferability further in environments with different characteristics (particularly with cities in more mountainous environments)".

**Response to referee #2**

**RC: I appreciate the efforts and kind responses of the authors. I agree with most of their modification.**

**The only left concern of mine is the validity of research data. As alternative models to physical models, they highly rely on physical models. Therefore, the accuracy of the physical model is the key factor to determine the effectiveness of flood forecasting. As the author said, the lack of monitoring data may bring great challenges to the verification of physical models. However, the multi-source data such as social media data and satellite monitoring may provide some support for the verification of physical models. If possible, add my concerns at the end of the manuscript.**

**In general, I think the paper has reached the publication levels of nhess.**

AR: We thank the referee for the positive feedback, and the recommendations. We added the following sentences to the manuscript "It is worth mentioning that the accuracy of the predicted flood maps by a data-driven model highly depends on the accuracy of the used hydrodynamic model simulations to train the model. While urban area lacks monitoring devices, crowd-sourced data and fine-resolution satellite images could be helpful tools to validate the hydrodynamic models."